# Microbial PolyHydroxyAlkanoate (PHA) Biopolymers—Intrinsically Natural

**DOI:** 10.3390/bioengineering10070855

**Published:** 2023-07-19

**Authors:** Anindya Mukherjee, Martin Koller

**Affiliations:** 1The Global Organization for PHA (GO!PHA), 12324 Hampton Way, Wake Forest, NC 27587, USA; anindya.mukherjee@gopha.org; 2PHAXTEC, Inc., 2 Davis Drive, Research Triangle Park, Durham, NC 27709, USA; 3Institute of Chemistry, University of Graz, NAWI Graz, Heinrichstrasse 28/IV, 8010 Graz, Austria

**Keywords:** biopolymers, green chemistry principles, natural polymers, polyhydroxyalkanoates

## Abstract

Global pollution from fossil plastics is one of the top environmental threats of our time. At their end-of-life phase, fossil plastics, through recycling, incineration, and disposal result in microplastic formation, elevated atmospheric CO_2_ levels, and the pollution of terrestrial and aquatic environments. Current regional, national, and global regulations are centered around banning plastic production and use and/or increasing recycling while ignoring efforts to rapidly replace fossil plastics through the use of alternatives, including those that occur in nature. In particular, this review demonstrates how microbial polyhydroxyalkanoates (PHAs), a class of intrinsically natural polymers, can successfully remedy the fossil and persistent plastic dilemma. PHAs are bio-based, biosynthesized, biocompatible, and biodegradable, and thus, domestically and industrially compostable. Therefore, they are an ideal replacement for the fossil plastics pollution dilemma, providing us with the benefits of fossil plastics and meeting all the requirements of a truly circular economy. PHA biopolyesters are natural and green materials in all stages of their life cycle. This review elaborates how the production, consumption, and end-of-life profile of PHAs are embedded in the current and topical, 12 Principles of Green Chemistry, which constitute the basis for sustainable product manufacturing. The time is right for a paradigm shift in plastic manufacturing, use, and disposal. Humankind needs alternatives to fossil plastics, which, as recalcitrant xenobiotics, contribute to the increasing deterioration of our planet. Natural PHA biopolyesters represent that paradigm shift.

## 1. Introduction

The increasing production and use of synthetic and recalcitrant polymeric materials of petrochemical origin for packaging, personal care, and other uses are currently responsible for several global threats: the accumulation of plastic waste in aqueous and terrestrial environments, the formation of highly recalcitrant, xenobiotic microplastic particles causing the death of marine animals, deteriorating ecosystems and negatively affecting human health, as recently demonstrated by studies confirming the presence of microplastic particles, even in human blood [1], placenta [2], and the digestive system [3]. Not all consumed microplastic is easily excreted by the body; it is estimated that microplastic intake by humans results to an irreversible accumulation of about 41 ng microplastic particles 1–10 µm in size per person until the age of 70 [4]. It is suspected that microplastics in our body may drive carcinogenic signaling [5]. Interestingly, in many environments, a so called “eco-corona” forms on the surface of microplastic particles, consisting of viruses, microbes, or organic toxins. This “eco-corona” facilitates the uptake of microplastic by human cells, where organisms and toxins unleash their toxic effects, and may even allow them to cross the blood–brain barrier [6].

In response to these threats, policies worldwide are striving to replace unsustainable linear economy approaches (“take-use-dispose”), like the production and use of fossil and persistent plastics, with circular alternatives (“take-use-recycle”). Legislations such as California Assembly Bill (AB) 1201 on “Solid waste: products: labelling: compostability and biodegradability” [7], Senate Bill (SB) 54 on “Solid waste: reporting, packaging, and plastic food service ware” [8], the European Commission’s Single Use Plastics Directive or SUPD (EU 904/2019) [9], the Plastics and Packaging Waster Directive (PPWR) [10], and the REACH Amendment on Intentionally Added Microplastics [11] go in this direction. The overarching goal of the current legislative efforts are to reduce virgin plastics use, their leakage into the environment, and the resulting pollution. In encouraging plastic recycling, policy makers seem to believe that the effects of persistent fossil plastics can be reduced or even eliminated. Recycling needs to be encouraged and improved; the sheer volume of fossil plastics used today (annual global production is estimated to be about 400 Mt and forecast to double by 2040) has no immediate replacement [12]. However, recycling only postpones the negative effects of fossil plastics by postponing the end of life of the materials. Chemical recycling is in its infancy, and mechanical recycling increases microplastics formation that are inherent in all plastics handling, processing, and use. These policies and legislations, while restricting fossil plastics, are also restricting the introduction of innovative alternatives. Policies and legislations are redefining words like “plastics” and “natural polymers” that are restricting the introduction of innovative, circular, and inherently sustainable alternatives. Instead, these policies need to focus on the circularity and sustainability of materials, through (a) renewable carbon use and from the (b) end-of life perspective of the material and not just the product, such as packaging or personal care items. Some examples of restrictive legislations have been mentioned above. Many of these legislations such as California SB 54 [8] and the European Union’s Single Use Plastics Directive (SUPD) [9] are focused on defining materials by redefining terminologies like “plastics” and “natural polymers” to restrict the use of these persistent fossil plastics with the aim of reducing harmful plastic pollution. Instead, we believe that legislation needs to focus on the circularity and end-of-life profiles and options, rather than focusing on redefining established terminologies. For example, renewability, biodegradability and compostability should be emphasized, along with standards and certifications that allow for such materials to be used. A second area that these legislations should focus on are encouraging and mandating increased separation and collection, and composting infrastructure to facilitate increased organic recycling. Examples of such legislations, which allow for biodegradable and compostable materials are California Assembly Bill 1080—“Solid waste: packaging and products” [13], or California SB 270, which imposes a statewide ban on single-use plastic bags at large retail stores [14]. Here, it should be noted that no nation-wide regulations exist for the US. Since 20 December 2022, the manufacture and import for sale in Canada of checkout bags, cutlery, foodservice ware, stir sticks, and straws (i.e., straight straws), as defined in respective regulations, are prohibited [15]. In the People’s Republic of China, a policy document jointly issued by the National Development and Reform Commission (NDRC) and the Ministry of Ecology and Environment on 16 January 2020, developed the plastic ban that prohibits restaurants throughout the country from providing single-use plastic straws and stores in the major cities from providing plastic shopping bags; this ban took effect on 1 January 2021 [16]. In April 2022, the Act on the Promotion of Resource Circulation for Plastics was enacted in Japan to improve the circularity of plastics [17]. In summer 2022, India imposed a ban on single-use plastic to tackle pollution, including straws, cutlery, ear buds, packaging films, and cigarette packets, etc. [18].

These legislations have also restricted materials that are renewable, biodegradable, and domestically and industrially compostable—attributes that would reduce and eventually eliminate the accumulation of persistent and fossil polymers and the dangers that they cause. Indeed, many of these legislations consider plastic recycling an appropriate strategy to overcome today’s plastic pollution predicament [19]; however, plastic recycling rather postpones the problem, and does not provide a solution. Instead, real “Natural Polymers” should be forcefully promoted by legislation. Widespread adaptation of natural alternatives is currently impeded by higher production costs, insecure supply chains for raw materials, or cumbersome downstream processing; however, natural polymers as solutions exist or are in development that use inexpensive raw materials, ecologically friendly polymer production and extraction methods, in addition to being circular and sustainable from an end-of-life perspective [20].

In this review, we clarify: (i) what “Natural Polymers” are, and (ii) how they comply with the term, plastic, in the truest sense of the word. In addition, we highlight attributes in materials that enable circularity and sustainability without the additional burden of fossil carbon use and plastics produced thereof.

## 2. Definition of “Natural Polymers”—Clarifying Existing Ambiguities

When it comes to sustainable and circular materials, nature is the best teacher. What nature makes, nature unwinds and remakes. Therefore, the prefix “bio” is frequently used to connect an object or a product to nature. Several such words exist:-**Bio-based**—Denoting a chemical compound or material produced from renewable carbon sources. “Bio-based” does not specify whether the product or material is synthesized chemically or biologically in plants, organisms, and animals.-**Biosynthesis**—The synthesis of a chemical or material through the actions in living organisms (plants, microorganisms, or animals) or parts thereof (enzymes). A biosynthesized polymer implies having been produced in plants, microorganisms, or animals through the action of enzymes in vivo. The word “polymerization” implies a synthetic or an anthropogenic process for generation of polymers from building blocks that are either bio-based, synthesized, or chemically synthesized.-**Biodegradation**—This implies the breakdown of a chemical or material through the actions of naturally occurring enzymes in vivo or by extracellular enzymes that are secreted by plants, microorganisms, or animals. Generally, breaking down chemicals or materials involves the eventual transformation to carbon dioxide (CO_2_) and water and complex biomass called humus, also known as organic fertilizer. Hence, “biodegradation” can be understood as nature’s way of material recycling.-**Biocompatibility**—Implies chemicals or materials that do not exert harmful effects on the environment or on living systems, including humans.

Materials that fulfil these criteria are well known, such as cellulose, starch, chitin, alginates, proteins, and nucleic acids (DNA, RNA). All these are also macromolecules and are excellent examples of “Natural Polymers”, also called a “Biopolymers”. Therefore, “Natural Polymers” are defined as those that are simultaneously:Derived from renewable carbon such as sugar from sugar cane, fatty acids like canola oil or carbon dioxide, or methane from biogas or the atmosphere.Biosynthesized in plants, microorganisms, and animals.Biodegradable or turn into CO_2_, water, and humus.Are biocompatible.

## 3. PolyHydroxyAlkanoates Are Natural Polymers

PolyHydroxyAlkanoates (PHAs) fulfil every criterion set out in the definition of a natural polymer or a biopolymer. PHA biopolymers, a group of biopolyesters, are found in nature, and they are biosynthesized using renewable carbon in microbes. PHAs are biodegradable, because nature has the tools to convert them into CO_2_ and water and about 10% organic fertilizer or humus, the same as in the biodegradation of cellulose or cotton. PHAs are biocompatible, they are harmless to all living beings, and can be used in vivo as medical devices such as scaffolds, stents or as meshes. They will eventually be absorbed by the body with no traces of the PHAs remaining [21].

### Evidence: PHAs Are a Natural Polymer

We spoke of nature knowing best when it comes to the circularity of chemicals and materials that are found in nature. For what nature creates, nature also develops ways to degrade and recycle. This is especially true for cellulose as well as PHAs. After their end of life, PHAs become mineralized or biodegraded by natural hydrolytic depolymerase enzymes to exactly those chemical building blocks from which they were once made in nature. These include CO_2_ and water, which then, through photosynthesis, are converted into sugar (sugarcane or corn) or lipids (canola oil or palm oil), used today to produce PHAs. This demonstrates that the formation and degradation of PHAs is inherently circular.

Figure 1 illustrates the fact that PHA biopolyesters are entirely embedded in nature’s closed material cycles, which is vividly exemplified by the old quote attributed to Lavoisier, a pioneer in the field of chemistry, when he explained that in nature “Nothing gets lost, nothing gets created, everything gets transformed.” [22].

Table 1 summarizes the aspects, which are integral to defining polymeric materials as “natural”, and how PHAs meet these requirements. In addition, Table 2 examines PHA manufacturing in the context of the 12 Principles of Green Chemistry, mentioned in individual sections of the text.

## 4. PHAs Are Produced by Natural Microorganisms

The production of PHAs occurs in naturally occurring prokaryotic microorganisms, which can be isolated from natural sources as diverse as the ocean, estuaries, soil, salt rocks in the mountains, plant surfaces, the rhizosphere, or the guts of insects. Hundreds of microbial species from both prokaryotic domains, Bacteria and Archaea, are reported to produce PHA. These microbial “biopolymer cell factories” thrive under diverse ecological conditions regarding their optima of temperature, pH-value, salinity, substrates, or concentrations of organic or inorganic toxins; hence, as well as mesophilic organisms, extremophiles are reported as PHA accumulators. The most comprehensively described PHA production strain, *Cupriavidus necator* [25], or *Bacillus* sp., in which PHAs were detected for the very first time by Maurice Lemoigne [26], were originally isolated from soil samples. Other strains, which are currently heavily studied due to their robust nature and convenient cultivability under low-sterility conditions, such as *Halomonas* sp., are marine isolates [27]. From polluted habitats like oilfields, versatile PHA-producing species like *Aneurinibacillus thermoaerophilus* have been isolated [28]. PHA biosynthesis has been observed in both heterotrophic (conversion of organic carbon sources for biomass, product and energy generation) and (photo)autotrophic (CO_2_ as inorganic carbon source) species isolated from diverse natural samples [29].

## 5. Biosynthesis of PHA in Natural Strains Occurs via Natural Biocatalysts

PHA biopolyesters are accumulated in natural organisms as intracellular granules; they serve the harboring cells as reserves for energy and carbon, electron sinks, and stress protectants, and provide PHA-containing cells with survival advantages under conditions of starvation, due to lacking an exogenous carbon source. They are predominantly produced by the cells under conditions of high intracellular energy charge, ample availability of exogenous natural carbon sources, combined with deprivation of additional growth factors such as nitrogen or phosphate sources. Such conditions inhibit the formation of catalytically active biomass but boost PHA productivity. Hence, PHAs are synthesized as typical intracellular products of a cell´s secondary metabolism [30]. As a precondition for PHA biosynthesis, a given strain needs to possess the key biocatalysts (enzymes) for PHA biosynthesis. The first is 3-ketothiolase (E.C. 2.3.1.9), which condenses two molecules of acetyl-CoA, the central catabolite of the breakdown of natural resources, to acetoacetyl-CoA that in turn gets reduced by the action of a reductase enzyme (E.C. 1.1.1.36), generating *R*-3-hydroxybutyryl-CoA, the substrate for PHA synthase (polymerase) enzymes (E.C. 2.3.1.304), which link the individual *R*-3-hydroxybutyryl-CoA building blocks to form polymeric PHA chains in vivo [31]. Therefore, the catalysts needed for PHA biosynthesis (the enzymes) match Principle 9 of Green Chemistry, which says: new catalysts shall be non-toxic, selective, and efficient [23].

This is vastly different from poly(lactic acid) (PLA), where polymerization of biologically produced lactic acid occurs via chemical processes (dimerization, ring opening polymerization), which requires synthetic, expensive, and often hazardous catalysts [32]. Importantly, PHA biosynthesis occurs under mild conditions, reflecting the optima of the strains’ enzymes for pH-value, temperature, salinity, or pressure. Production of petroplastics, on the contrary, needs harsh, energy-demanding reaction conditions, and typically expensive and hazardous catalysts. As an illustration, the production of poly(ethylene terephthalate) (PET, a polyester) requires temperatures above 200 °C, and typically requires antimony oxide as a catalyst [33]. Thus, PHA biosynthesis follows Principle 3 of Green Chemistry, which demands safe chemical reactions for the production of safe products, from safe educts [22].

## 6. PHA Biosynthesis Is Based on Natural Feedstocks

Principle 7 of Green Chemistry postulates the replacement of fossil feedstocks by renewables [23]. This is perfectly valid for PHAs; carbohydrates stemming from the photosynthetic fixation of CO_2_ are the predominant natural feedstocks used for PHA biosynthesis (Figure 1). This goes especially for monosaccharides such as the hexoses glucose or fructose and, for a lower number of microorganisms, the pentose xylose. These monomeric sugars can conveniently be generated from polysaccharides like starch, cellulose, or even abundantly available lignocellulose via well-established hydrolysis techniques (enzymatic or chemical). Notably, these polysaccharides frequently stem from agro-industrial waste; upcycling of such waste to feedstocks integrates PHA biosynthesis into biorefinery concepts, which perfectly matches current circular bioeconomy paradigms. In many cases, hydrolysis of said polysaccharides is not even needed; a cohort of strains is reported to directly convert di-, oligo-, and polysaccharides to biomass and PHAs, provided the strains have the adequate hydrolase enzymes available. Examples include the expedient direct starch converter *Haloferax mediterranei*, which has high amylolytic activity [34]; the invertase-excreting strain *Paraburkholderia sacchari* [35], which readily converts the disaccharide sucrose; or *Hydrogenophaga pseudopflava*, a well-described consumer of the disaccharide lactose [36]. Other strains are excellent PHA accumulators from lipids produced by natural organisms (microbes, plants, or animals), such as *Pseudomonas putida* [37]. Emerging PHA production processes are based on gaseous substrates; here, a range of hydrogen oxidizers like *C. necator* [38] and many phototrophic cyanobacteria [39] can grow and accumulate PHAs by using CO_2_ as the sole carbon source; CO_2_, in turn, is the final product of the bio-mediated aerobic breakdown of biomass and also of disposed items made of PHAs. Moreover, several Type II methanotrophic bacteria can be cultivated on CH_4_, a product of the composting and anaerobic degradation of natural products like organic waste and PHAs. It is also the main component of natural gas [40]. This again nicely illustrates the circularity of PHAs: After their life span, bioplastic products made of PHAs undergo natural degradation to exactly those starting materials, which, in a future cycle, fuel PHA biosynthesis (Figure 1). Finally, the microorganism class of Rhodospirilli are known to produce PHAs even from CO-rich syngas, a substrate accessible from pyrolysis of most biological resources, particularly waste biomass [41].

## 7. PHAs Are Biodegraded in Nature—Marine, Fresh Water, and Soil in Every Environment around the World

Principle 10 of Green Chemistry is about “biodegradability”: Products should be able to degrade naturally (or “biodegrade”) after use without harming the environment [23]. As has been mentioned above, aerobic (oxidative) degradation of PHAs by microbes like bacteria or fungi generates CO_2_ and water, while anaerobic PHA consumption by living organisms, e.g., in biogas plants, results in the generation of CH_4_ in addition to water and CO_2_. While the biodegradability of PHAs has been long established, decisive factors influencing biodegradability of PHA, such as shape and thickness of polymer specimens, crystallinity, composition on the level of monomers, environmental factors (humidity, pH-value, temperature, UV-radiation), and surrounding microflora, though comprehensively studied and reviewed, are currently under scrutiny. While more studies are needed to further refine biodegradation timelines for various types of PHAs, the fundamental fact that those types of PHAs produced to date at reasonable quantities are biodegradable has already been established [42]. This variability in the biodegradation of PHAs follows all other natural materials including starch, cellulose, proteins, or chitin.

Biodegradability and compostability of PHA biopolyesters have been scrutinized under diverse environments and test conditions, i.e., soil, water, marine, simulated body fluids, activated sludge, as well as industrial and domestic composting. PHA-producing companies have tested their PHA products according to the corresponding standards of certain certification organizations in order to verify claims of biodegradability and compostability of each product [43,44,45,46]. Standards and specifications have been developed by several authorities such as the European Committee for Standardization (EN), the American Society for Testing and Material (ASTM), the International Organization for Standardization (ISO), the British Standard Institution (BSI), etc.

Even the highly crystalline poly(3-hydroxybutyrate) (P(3HB)) homopolymers, the most studied type of PHA, are biodegradable and compostable in nature. The P(3HB) producer, Biomer, has confirmed that P(3HB) is “fully biodegradable” and compostable [47]. Previously, commercially available P(3HB) homopolyester from Imperial Chemical Industries (ICI), UK, even exceeded the biodegradability of Novamont´s thermoplastic-starch (TPS)-based composite material, Mater-Bi. At different temperatures (28, 37, and 60 °C), the biodegradability of these materials has been studied by burying thin sheets made of them in forest soil, sandy soil, activated sludge soil, and in farm soil. P(3HB) showed almost 100% degradation (average mass loss of five parallel samples: 98.9%) in activated sludge soil at 37 °C after only 25 days, while degradation in farm soil (68.8% mass loss after 25 days) sandy soil (10% mass loss), and forest soil (7% mass loss) was considerably slower, again showing the high impact of the environment and microflora on PHA biodegradability. Remarkably, no complete degradation was achieved for TPS (72.1% degradation in activated sludge after 25 days at 60 °C as the highest degradability) at any condition studied [43]. Regarding PHA biodegradability in seawater, the degradability of P(3HB), poly(3-hydroxybutyrate-*co*-3-hydroxyvalerate) (P(3HB-*co*-3HV)), and poly(3-hydroxybutyrate-*co*-4-hydroxybutyrate) (P(3HB-*co*-4HB)), films were studied in a marine environment over one year. It was shown in this study that degradation rates strongly depend on the temperature of seawater; moreover, mechanistically, all PHA samples were degraded via surface erosion, where depolymerase enzymes first attack the amorphous P(3HB) regions on the surface of a polymer sample; after that, crystalline P(3HB) regions are depolymerized [48].

Finally, PHA biopolymers of different structures and compositions (homo- and copolyesters) can be mixed with other natural, compatible materials, such as PLA, wood dust, lignin, etc. [49]. This generates biocomposites and blends with tailor-made properties, such as optimized gas barrier properties, which makes these products more functional, e.g., for food packaging purposes while maintaining their compostability characteristics at their end-of-life [50]. Often, inexpensive, bio-based agro-industrial residues, such as bagasse straw, are incorporated into PHA matrices, which reduces the overall production cost and the density of the biopolymer items, while maintaining PHAs’ desired performance in given applications, and, at the same time, does not impede biodegradability. Finally, the composition of such biocomposites is being fine-tuned so that they can be processed using existing equipment and techniques, such as injection molding or film blowing, which can be challenging when using pristine PHAs [51].

## 8. PHAs Do Not Create Recalcitrant Microplastics

Typically, fossil plastics that are disposed as macro-plastic waste in landfills and in the environment, turn into micro- and nanoplastic particles (“secondary microplastic”) sized between 1 nm and 5000 µm through erosion and abrasion. Such nano- and microplastic particles are also generated during the use of the fossil plastics by abrasion during diverse industrial processes, from vehicle tires, and even from shoe soles. [52,53]. Nanoplastic and microplastic particles have the appropriate size to enter the food chain starting with plankton which ingest them. Plankton serve to nourish higher animals like fish, and the nanoplastic and microplastic particles finally end up on our table [54]. Plastic recycling, a strategy often promoted by legislative regulations, is a key source of microplastic formation. This was clearly shown by a study on mineral water sold in PET-bottles: Recycled bottles released tremendously more microplastic particles into the water than bottles made of virgin PET did [55].

At this point, it needs to be emphasized that small-sized PHA particles are not resistant in nature; they undergo biodegradation and do not leave any remnants. Hence, “secondary microplastics” consisting of PHA biopolymers simply do not exist. This matches the first principle of Green Chemistry, which postulates that the generation of precarious waste shall be avoided; it is “better to prevent than to cure” toxic waste like fossil microplastic, which can be achieved by switching to PHAs [23]. The circularity of PHA biopolyesters is already being commercially exploited for many uses, including for replacing intentionally added fossil microplastics (“primary microplastic”) in cosmetics to offer UV protection (sun screen), and in skin peeling and scrubbing products [56]. Intentionally added, PHA microparticles that are added in shower gels, cosmetic peeling and scrubbing agents are released into the environment such as in sewage and waste water treatment plants, lakes, rivers and in the marine environment undergo biodegradation, unlike fossil microparticles, which remain recalcitrant. These persistent microplastics cause excessive microplastic loads in sewage sludge generated in wastewater treatment plants. Wastewater treatment plants filter up to 99% of the microplastic from domestic wastewater, but with PHA microparticles this would not be needed. The presence of PHA microparticles in sewage and wastewater treatment plants have an added benefit. PHAs, while undergoing biodegradation convert the high-nitrogen-content chemicals such as nitrates and nitrites present in sewage sludge into nitrogen gas, a process also called “denitrification”, thus aiding in improved sewage treatment [57]. Sewage sludge is an excellent organic fertilizer; however, currently, they contain significant microplastics which end up in agricultural fields when used as fertilizer [58]. Replacing fossil microparticles with natural PHA microparticles would allow the PHA microparticles in sewage sludge to readily biodegrade in the sewage system or in the fields by the microflora present there.

Products containing natural PHA microparticles have already been commercialized— such as Naturetics^TM^ products by the Nafigate Cooperation, which was launched in the Czech Republic in 2021—and are based on Hydal technology, to produce PHAs from waste cooking oil [59]. In the most recent Microplastics Amendment (to Annex XVII) within the REACH Regulation, the European Union has again introduced language that restricts the use of biodegradable materials by tying the amendment to a definition of “Natural Materials” that the European Union (ECHA) created within the SUP Directive (EU 2019/904) [9]. Even though under the new amendment, any new intentionally added microplastic must meet specific biodegradability standards, one of those positive controls (standards) is P(3HB), a type of PHA biopolymer. This type of action by the European Union is being picked up by other nations, creating a patchwork of legislations that are negatively affecting investments, commercialization, and the adoption of these molecules that nature created and that humans have refined to make and break down in a circular way.

## 9. Industrial PHA Production Is Analogous to PHAs Produced in Nature

PHAs are produced by microorganisms in vivo, as an energy source that the organism itself utilizes to survive in nature when it cannot obtain an external carbon source [60]. Hence, PHA production can be compared to plants producing cellulose when they grow, although cellulose is a structural part of the plant, not a reserve compound like PHAs. On an industrial scale, manufacturers use the same natural microorganisms to produce PHAs, the industrial process being referred to as “fermentation”. Many of our food products are manufactured using microorganisms such as grapes are fermented to produce wine, milk is fermented to produce cheese and yogurt, and cabbage transforms into sauerkraut through fermentation. PHA manufacturers also utilize the same natural process of renewable carbon substrate uptake by their microorganism of choice to produce PHAs, i.e., the microorganisms carry out their natural processes in adequate vessels or containers, called “bioreactors” or “fermenters”. On an industrial scale, manufacturers produce PHAs using microorganisms of their choice under controlled cultivation just like in the case of producing fermented food products. They do so in “bioreactors” or “fermenters” of different sizes and configurations, where cultivation conditions (temperature, pH-value, dissolved oxygen tension, actual substrate concentration, etc.) have been optimized through research and development to produce PHA biopolymers economically [61]. Prime examples of similar processes are the aerobic propagation of baker’s yeast biomass for making bread [62], high-throughput vinegar manufacturing in well-aerated bioreactors (“acetators”) [63], the production of penicillin antibiotics by *Penicillium* sp. [64], or citric acid production by *Aspergillus niger* [65]. Specifically, penicillin and citric acid production processes are based on submerged microorganism cultivations in bioreactors similar to PHA production; in all three cases, two cultivation phases are observed: first, high densities of active biomass are produced under nutrient-rich conditions, which, provoked by the limitation of a growth-essential nutrient, generates the desired bioproducts in a second cultivation phase as typical secondary metabolites. The two processes run sequentially, typically in the same bioreactor. The only major difference is that penicillin and citric acid are excreted by the production strains as extracellular products [64,65], while PHAs are stored in cells intracellularly and need to be released from the cells. The recovery of PHAs from cells is increasing based on the application of natural techniques, such as enzymatic disintegration of non-PHA biomass, or the application of biogenic “green” extraction solvents [66]. Coming back to the 12 Principles of Green Chemistry, industrial PHA production adheres to Principle 8 (reduction of intermediate production stages and by-product formation, efficient and intensified processes [23]), Principle 11 (real-time process monitoring using modern bioreactor equipment, in-line analytics, and digitalization [23]), and Principle 12 (general risk mitigation through the selection of safe raw materials, safe process management, and the avoidance of explosions, fires, and the release of toxins [23]). Moreover, Principle 6 talks about energy efficiency in manufacturing processes: Processes shall be carried out under mild process conditions, such as at room temperature and atmospheric pressure [23]. This is exactly the case for biotechnological processes like PHA production, which take place at the biologically optimum, mild conditions for a given production strain. Finally, Principle 5 refers to the use of safe and non-toxic solvents in chemical processes [23]: The solvent used in bioreactors for PHA production is the safest solvent imaginable: water.

## 10. Products Made of PHAs Are Natural and Biocompatible

PHAs can be processed with existing machinery to vendible biopolymeric products of high utility, e.g., via injection molding, blow molding, extrusion molding, compression molding, additive manufacturing (3D-printing), electrospinning, etc. [67]. During melt processing, PHAs retain their basic chemical structure and are not modified, such as with crosslinked or thermoset materials [68]. Hence, PHAs undergo no chemical modification from the time they are biosynthesized to its melt forming, except that heat treatment of all polymers generally lowers their molecular weight. However, the end-of-life scenarios of products made of biopolymers like PHAs differ from their fossil competitors: according to Peng et al., biodegradable polymers such as PHAs are materials that can work for a limited time before degrading into “readily discarded products” (end products of mineralization) through a “regulated procedure” (enzymatic biodegradation) [69]. Hence, the circularity of products made of PHAs is completed through their end-of-life options: they are biodegradable both under aerobic conditions (composting in both domestic and industrial conditions) and without oxygen (biogas and anerobic digestion plants). Therefore, no persistent macro- and microplastic waste remains from leftovers of PHA-based materials, in contrast with fossil plastics, which are persistent. Again, this meets the definition of “Natural Polymers”.

This pronounced propensity to biodegradability is illustrated by numerous biodegradation studies carried out worldwide in academic and industrial laboratories, and at the laboratories of certification bodies. The most diverse types of PHAs produced in small quantities at research laboratories, as well as on a large scale by various companies, have already been studied to assess their biodegradability. These industrially produced types of PHAs include the homopolyester P(3HB) and the copolyesters of *R*-3-hydroxybutyrate (3HB) and the co-monomers, *R*-3-hydroxyvalerate—3HV, 4-hydroxybutyrate—4HB, or *R*-3-hydroxyhexanoate—3HHx). All these materials, although being of different crystallinity and processability, have been shown to be readily biodegradable under diverse test conditions, such as at various temperatures and in soil, fresh water, or marine environments [70]. Importantly, for several types of PHA, biodegradation, hence, the conversion of the carbon in the PHAs into CO_2_ outperformed the biodegradation performance of cellulose, the prototype natural reference material. Remarkably, the compostability of PHAs, namely, its disintegration into tiny fragments, which is essentially the recycling of carbon, not only occurs under industrial compositing conditions characterized by elevated temperature, but also with “bioplastics” like PLA; beyond that, PHAs can even be composted at home, which allows for the simple and efficient management of polymeric household waste, when considering the disposal of PHA-based packaging materials [70].

Uses of PHAs: PHAs are suitable for replacing fossil plastics in packaging, personal care, and agriculture. There is a significant effort underway to use PHAs in single-use packaging, where recycling packaging materials is difficult due to their complex multi-material construction and light weight. In foodservice packaging, contamination of the packaging with food makes recycling even more difficult. Short-shelf-life (days, weeks, or months) foodservice packaging applications are ideal for a biodegradable and compostable material like PHAs. Their excellent barrier properties to gases, moisture, and fats and oils add to the advantages they offer in these applications.

However, PHAs can also be applied in long shelf-life uses such as in personal care packaging for soap and shampoo bottles that requires longer product shelf life. It must be emphasized that in long-shelf-life packaging and other durable products, PHAs do not biodegrade, and keep their integrity and functionality just like fossil plastic bottles. Biodegradation starts when PHAs come in contact for long periods of time with adequate microflora, e.g., in composting facilities, in soil, or in water. Ultimately, the biocompatibility of PHAs constitutes the final “brick in the wall” built by the rest of the attributes, demonstrating that PHAs are natural. This is shown by the huge number of eco- and bio-toxicity tests carried out with PHAs, and their successful application as drug carriers, scaffold materials, nerve-repair conduits, artificial blood vessels, surgical pins and wires, and stents. In vivo applications of PHA-based implants do not exert inflammation reactions, because PHAs are metabolized in tissue to metabolites constitutively present in the human body, such as 3HB. This again is in contrast to PLA-based implants, which can cause inflammation due to lactic acid formation [21]. Most of all, it should be noted that resorbable surgical meshes made of poly(4HB) and P(3HB-*co*-4HB) are FDA approved, and have already been successfully commercialized by the company Tepha, Inc., owned by BD (Becton, Dickinson and Company) since 2021. These materials constitute macro-porous, monofilament, fully absorbable scaffolds with long-term strength retention and a defined absorption time [71]. The biocompatibility of PHAs adheres to Principle 4 of Green Chemistry, which is about the development of safe products, which, while exhibiting beneficial performance, are of no or minimal toxicity [23].

PHAs are demonstrably biodegradable and compostable when subjected to sustained enzymatic or microbial treatment; however, as materials in use, they are extremely durable in numerous applications. Just as cellulosic papyrus and old scrolls have survived many millennia when appropriately preserved, PHA articles would be long-lasting, until they undergo sustained enzymatic or microbial action in nature, at wastewater/sewage treatment plants, or during industrial composting. This durable attribute of PHAs makes them attractive as a replacement for many fossil plastics.

## 11. Definition of Plastics and How PHAs Are Plastics

Defining “plastics”: The term “plastic” derives from the Greek word, πλαστικός (*plastikos*), which means “can be shaped or molded”, while πλαστός (*plastos*) means “molded”. Additionally, “plasticity” explicitly refers to the deformability of materials, which in this case, refers to the polymers used in plastic manufacturing. A plasticity of materials enables molding, extrusion, or compression into specimens of various shapes: films, fibers, meshes, plates, bottles, tubes, containers, etc. Plasticity also has a technical definition in materials science, referring to the non-reversible change in form of solid substances, such as in the case of thermoplastics that are materials that can be shaped through the application of heat and pressure but retain their shape at ambient or lower temperatures below their melting point.

This definition of thermoplastics has been transported to popular culture and vocabulary from the scientific vocabulary over the last 50 years to refer to polymers made from fossil carbon. The reference has a rationale: Almost all polymers that can be shaped like thermoplastics have originated from fossil plastics since the 1950s. Therefore, “plastics” today generally refer to fossil-derived plastics; however, if we refer to the original definition of plastics, meaning a material that can be shaped and reshaped using heat and pressure, PHAs would fall in that definition.

What makes PHAs “plastics”: PHA polymers are also thermoplastics as described above; they can be processed on conventional processing equipment in which fossil plastics are melt processed to manufacture marketable products.

This thermoplastic characteristic of PHAs is an important distinction relative to other biopolymers such as cellulose. It is this characteristic of PHAs that allows them to be processed like fossil plastics. PHAs are a class of biopolymers with around 150 different building blocks, allowing for numerous combinations of polymers with properties that cover a significant range of properties that mimic the properties of the top seven best-selling fossil plastics, including polyolefins such as polyethylene, polypropylene, polystyrene, or poly(vinyl chloride), and polyesters such as PET [30]. They differ in their properties which are dependent on their chemical composition (homo-or copolyesters, the type of hydroxy alkanoic acids making up the biopolyester); PHAs made up of building blocks with three to five carbon atoms (short-chain-length PHA) are typically thermoplastics with amorphous and crystalline moieties (up to 70% crystallinity), possessing melting temperatures ranging from 100 °C up to 180 °C [20]. The most described PHA, the thermoplastic homopolyester P(3HB), is similar in its material properties to PP, has good resistance to moisture, and excellent gas and aroma barrier properties; it can be used in applications where high hardness or creep-resistance are needed. P(3HB) does not change their properties over a broad temperature range even when stored for several years. The UV-resistance and mechanical stability of P(3HB) even outperforms many competing petrochemical thermoplastics [72]. Processability, impact strength and flexibility of PHA grades improve with the introduction of co-monomers such as 3-hydroxyvalerate, 4-hydroxybutyrate, or 3-hydroxyhexanoate. Such PHA co-polymers or “copolyesters” are already being commercially produced [73]. In contrast to short-chain-length PHAs, those PHAs, consisting of building blocks with at least six carbon atoms (medium-chain-length PHAs), are more amorphous, have a low melting temperature, and display elastomeric properties similar to latex [20]. Industrial PHA biopolymers are already being used in packaging applications where food contact and contamination renders recycling difficult and such products are easily composted [73]. Once PHA volumes increase, they could also be recycled like fossil plastics, due to their thermoplastic nature, thus making PHAs recyclable, as well domestically and industrially compostable [70].

## 12. Conclusions

This technical review has presented the criteria which need to be considered in classifying polymeric materials as natural materials or natural polymers. Nature uses renewable carbon as a raw material and nature’s workhorses (enzymes in living organisms) utilize biosynthesis to produce natural materials or natural polymers. In addition, nature has the tools to unzip or biodegrade them. PHAs are one of those natural materials or natural polymers. They are products of natural living organisms produced from renewable carbon raw materials. Their chemical structure remains unchanged during or after melt processing them into diverse biocompatible consumables that do no harm to living systems, and they do not disintegrate into persistent microparticles. Their beneficial end-of-life options, namely recyclability and biodegradation via composting or anaerobic digestion to biogas, are analogous to all other known biopolymers, such as carbohydrates, proteins, or nucleic acids.

Accelerating the transition to biological alternatives like PHAs on a substantial scale requires a paradigm shift in many areas. This shift would involve the availability of a variety of non-food renewable carbon sources and ecologically benign biopolymer recovery methods; designing materials that match existing fossil plastics in their properties and functionality, enabling their processing on existing infrastructure with little to no changes; designing waste management systems that improve collection and sorting; and accepting and improving industrial composting and making it equally as important as mechanical recycling. In addition, appropriate legislation to promote their use needs to focus on circularity and sustainability—renewability, recyclability, compostability, and biodegradability. It must not focus on redefining plastics, natural polymers, and other clearly established words and phrases. FMCGs (fast-moving consumer goods) and CPGs (consumer packaged goods), especially in food service, personal care, and health and beauty are ideally suited for such a transition to biological alternatives like PHAs [73].

In addition, the overall cost of using fossil plastics that includes their negative environmental consequences needs to be holistically estimated beyond the cost of their raw materials. This would allow for a true understanding and comparison of costs of fossil plastics with their biological and circular alternatives.

Finally, governments need to provide incentives in the form of direct support for more basic and applied research and development including the scaling newer technologies to enable the transition. This area represents the “New Economy”, also called the “Bio-Economy”, allowing local supply chains for materials, and new and innovative manufacturing processes that create significant new employment opportunities.

## Figures and Tables

**Figure 1 bioengineering-10-00855-f001:**
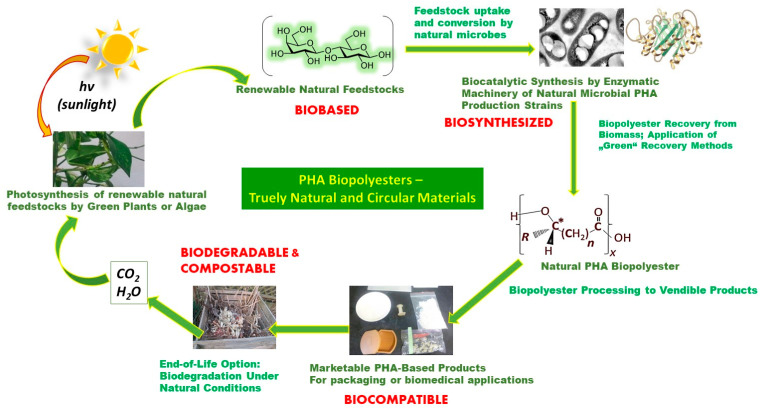
Circularity of natural PHA biopolyesters.

**Table 1 bioengineering-10-00855-t001:** Comparison of different natural/non-natural polymers.

Material:	PHA	Carbohydrates (Starch, Cellulose, Alginates, Chitin, etc.)	Proteins	PLA	PCL	Bio-PE	Fossil Plastics (PE, PP, PVC, PET, etc.)
Criterion:							
Production depletes fossil resources	NO	NO	NO	NO	YES	NO	YES
Biobased (produced from renewable natural resources or via photosynthesis)	YES	YES	YES	YES	NO	YES	NO
Biosynthesized (polymerization taking place in nature by nature´s enzymatic toolbox)	YES	YES	YES	NO	NO	NO	NO
Biodegradable (degradation in nature by enzymes present in living organisms)	YES	YES	YES	YES	YES	NO	NO
Biocompatible (not harmful to the biosphere)	YES	YES	YES	NO	YES	NO	NO
(Natural) chemical structure after (bio)synthesis can be maintained during processing	YES	YES	YES	YES	YES	YES	YES
Formation of persistent microparticles	NO	NO	NO	NO	NO	YES	YES
Incineration generates surplus atmospheric CO_2_, thus fueling global warming	NO	NO	NO	NO	YES	NO	YES
Summary: **Natural polymer or not??**	**YES**	**YES**	**YES**	**NO**	**NO**	**NO**	**NO**

PCL: poly(ε-caprolactone); PE: poly(ethene); PP: poly(propene); PVC: poly(vinyl chloride); PET: poly(ethylene terephthalate).

**Table 2 bioengineering-10-00855-t002:** PHAs in relation to the 12 Principles of Green Chemistry [23,24].

Principle	Description	Relevant to PHAs?	Explanation of the Relevance
**Principle 1:** **Prevention**	“It is better to prevent waste than to treat or clean up waste after it has been created.”	**YES**	Neither PHA biosynthesis nor PHA biodegradation generates any waste which needs to be treated, in contrast to production and disintegration of fossil plastics. This is especially true for the formation of microplastic particles, which, in the case of fossil plastics, are recalcitrant and endanger the eco- and biosphere, while micro-sized PHA particles undergo biodegradation.
**Principle 2:** **Atom Economy**	“Synthetic methods should be designed to maximize incorporation of all materials used in the process into the final product.”	**-**	Not applicable in the context of PHA. As natural, aerobic process, PHA biosynthesis creates CO_2_ as side product; however, generated CO_2_ is embedded into the natural carbon cycle, in contrast to CO_2_ stemming from petrochemical products.
**Principle 3:** **Less Hazardous Chemical Syntheses**	“Wherever practicable, synthetic methods should be designed to use and generate substances that possess little or no toxicity to human health and the environment.”	**YES**	PHA biopolyesters are highly biocompatible; they do not exert any risk to the environment or human health.
**Principle 4:** **Designing Safer Chemicals**	“Chemical products should be designed to preserve efficacy of function while reducing toxicity.”	**YES**	PHA biopolyesters can function as a replacement for plastics, but, in contrast to fossil plastics, do not generate any toxic compounds when disposed or recycled.
**Principle 5:** **Safer Solvents and Auxiliaries**	“The use of auxiliary substances (e.g., solvents, separation agents, etc.) should be made unnecessary wherever possible and, innocuous when used.”	**YES**	PHA biosynthesis occurs in the aqueous phase; hence, water, being the most sustainable and innocuous solvent, is used. For PHA recovery from biomass, ecologically benign “green solvents” of natural origin can be used.
**Principle 6:** **Design for Energy Efficiency**	“Energy requirements should be recognized for their environmental and economic impacts and should be minimized. Synthetic methods should be conducted at ambient temperature and pressure.”	**YES**	PHA biosynthesis typically occurs at room temperature and under ambient pressure conditions, which makes it a process of low energy requirements. Required energy supply is mainly due to upstream processing (e.g., sterilization of the bioreactor and media compounds), aerating, and stirring of the bioreactor, and downstream processing. In the case of using extremophilic organisms, energy for sterilization can even be avoided.
**Principle 7:** **Use of Renewable Feedstocks**	“A raw material or feedstock should be renewable rather than depleting whenever technically and economically practicable.”	**YES**	PHAs originate from renewable feedstocks.
**Principle 8:** **Reduce Derivatives**	“Unnecessary derivatization (use of blocking groups, protection/deprotection, temporary modification of physical/chemical processes) should be minimized or avoided if possible, because such steps require additional reagents and can generate waste.”	**YES**	Derivatization is not needed during PHA biosynthesis due to the efficiently coordinated enzymatic sequence in the intracellular biocatalytic cascade.
**Principle 9:** **Catalysis**	“Catalytic reagents (as selective as possible) are superior to stoichiometric reagents.”	**YES**	A highly efficient enzymatic cascade, starting from the biocatalysts responsible for substrate catabolism towards acetyl-CoA, propionyl-CoA, etc. (glycolysis, KDPG pathway, β-oxidation, oxidative pyruvate decarboxylation, etc.), via enzymes generating the PHA building blocks (3-ketothiolase and reductase) until PHA synthases catalyze PHA biosynthesis. Activities of enzymes involved in the entire pathway from substrate to bioproduct (PHA) are highly specific, and efficiently coordinated.
**Principle 10:** **Design for Degradation**	“Chemical products should be designed so that at the end of their function they break down into innocuous degradation products and do not persist in the environment.”	**YES**	Products made of PHAs degrade naturally after use, without harming the environment, into exactly those innocuous compounds they derive from: water and CO_2_ under aerobic conditions (composting), plus methane under anaerobic conditions (biogas plants).
**Principle 11:** **Real-time analysis for Pollution Prevention**	“Analytical methodologies need to be further developed to allow for real-time, in-process monitoring and control prior to the formation of hazardous substances.”	**YES**	Modern bioreactor equipment used for PHA production encompasses online analytical tools to monitor temperature, pH-value, dissolved oxygen concentration, redox potential, foam formation (via conductivity sensors), substrate consumption (e.g., glucose sensors), biomass formation (via turbidity sensors), and the CO_2_ level and remaining oxygen in exhaust gas. These parameters are processed via high-level digitalization, which control the process in real time by adjusting substrate supply, oxygen input, temperature, pH-value, etc.
**Principle 12:** **Inherently Safer Chemistry for Accident Prevention**	“Substances and the form of a substance used in a chemical process should be chosen to minimize the potential for chemical accidents, including releases, explosions, and fires.”	**YES**	PHA production is based on safe, renewable feedstocks. No risk of fire or explosion exists during PHA production, and no toxins are released during the process.

## Data Availability

Not applicable.

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
