# Peer review of "Microbial PolyHydroxyAlkanoate (PHA) Biopolymers—Intrinsically Natural"

_bioengineering, 2023, doi:10.3390/bioengineering10070855_

Round 1

Reviewer 1 Report

Even if I do not consider the manuscript as a Technical Review, as reported, because it lacks the main points to be considered like that. Anyway, I believe the manuscript could be interesting for the readers of Bioengineering. The manuscript provides a general overview about PHA production, consumption, end-of -life together with definition of key concepts like biodegradability, biosynthesis, natural polymers and others. Moreover, comparison among synthetic and natural polymers are reported. 

I suggest for publication, my only suggestion is to add some representative figures of the topic to make the manuscript more attractive

Author Response

Even if I do not consider the manuscript as a Technical Review, as reported, because it lacks the main points to be considered like that.

Answer: We changed to “Review” as article type

Anyway, I believe the manuscript could be interesting for the readers of Bioengineering. The manuscript provides a general overview about PHA production, consumption, end-of -life together with definition of key concepts like biodegradability, biosynthesis, natural polymers and others. Moreover, comparison among synthetic and natural polymers are reported.

Answer; Thank you for these positive and motivating comments!

I suggest for publication, my only suggestion is to add some representative figures of the topic to make the manuscript more attractive

Answer: We now provided a more detailed graphical abstract now to clarify the concepts of the paper. Additional graphics would exceed to aspired length of the manuscript. Central statements are, beside in Figure 1 and the Graphical Abstract, provided in the 2 Tables.

Reviewer 2 Report

It is an  interesting technical review focused on a very relevant topic that clearly needs to be clarified and in that form this manuscript will help to learn more and understand better about natural polymers, more specifically PHA. 

It is very well written and the subject is addressed in a different form,  such as the PHA contextualization on the Green Chemistry Principles.

This review come in the following of other works presented in this field by the authors. The only aspect that I would also like to see addressed would be the biodegradability of PHA when mixed with other natural polymers, since for applications as packaging the utilization of PHA as single polymer is not possible since it is not easy the injection molding or blow molding of these biopolymers.

In general the manuscript addresses a very captivating subject and can be helpful for people working with polymers and natural polymers.

Author Response

It is an  interesting technical review focused on a very relevant topic that clearly needs to be clarified and in that form this manuscript will help to learn more and understand better about natural polymers, more specifically PHA. 

It is very well written and the subject is addressed in a different form,  such as the PHA contextualization on the Green Chemistry Principles.

Answer; Thank you for these positive and motivating comments!

This review come in the following of other works presented in this field by the authors. The only aspect that I would also like to see addressed would be the biodegradability of PHA when mixed with other natural polymers, since for applications as packaging the utilization of PHA as single polymer is not possible since it is not easy the injection molding or blow molding of these biopolymers.

Answer: This section was added: Finally, PHA biopolymers of different structures and composition (homo- and copolyesters) can be mixed with other natural, compatible materials, such as PLA, wood dust, lignin, etc. [49]. This generates bio-composites and blends with tailor-made properties, such as optimized gas barrier properties, which makes these products more functional, e.g., for food packaging purposes while maintaining their compostability characteristics at end-of-life [50]. Often, inexpensive bio-based agro-industrial residues, such as bagasse straw, are incorporated into PHA matrices, which reduces the overall production cost and the density of the biopolymer items, while maintaining PHA´s desired performance in given applications, and, at the same time, does not impede biodegradability. Finally, the composition of such bio-composites are being fine-tuned such that they can be processed in existing equipment and techniques, such as injection molding or film blowing, which can be challenging when using pristine PHA [51].

In general the manuscript addresses a very captivating subject and can be helpful for people working with polymers and natural polymers.

Answer; Thank you for these positive and motivating comments!

Reviewer 3 Report

The manuscript is a technical review that elaborates in detail how PHA’s production, consumption, and end-of-life profile are perfectly embedded in the current topical 12 Principles of Green Chemistry, which constitute the basis for sustainable product manufacturing. 

1. How do microplastic particles negatively affect human health? Could you provide some references?

2. What is the source of the statistics that the annual production of fossil plastics is about 400 Mt? Is it worldwide or in the US only? Is this number stable over the years or growing? How it was affected during the Covid-19 pandemic?

3. What are the primary barriers to replacing fossil-based plastics with biodegradable alternatives?

4. Could you comment on the functionality and longevity of PHA-based materials in comparison with fossil plastics?

5. Regarding the Green Chemistry Principle No 2 Atomic economy, it does not fit into the context of PHA. How would you compare the impact of CO2 emission by comparing with the synthesis of fossil plastics?

6. I would suggest shortening the conclusions part and moving tables 1 and 2 to the discussion part.

 7. The issues of fossil plastics consumption and pollution are complex and require comprehensive solutions that involve scientific advancements, policy changes, and societal shifts. Future direction’s part should be added with recommendations for future research.

8. In technical and scientific writing, the use exclamation mark should be used only to end warning or caution statements or as specialized scientific notation. For other purposes, I would suggest using a period or question mark.

Author Response

  1. How do microplastic particles negatively affect human health? Could you provide some references?

Five new references added; risks of carcinogenic signaling and confirmed presence of microplastic in human blood, placenta, and the digestive system added. Moreover, the formation of toxic “eco-corona” on the surface of microplastic particles is added now, which shuttles toxins into human cells, and even allows microplastic coring the blood-brain barrier.

  1. What is the source of the statistics that the annual production of fossil plastics is about 400 Mt? Is it worldwide or in the US only? Is this number stable over the years or growing? How it was affected during the Covid-19 pandemic?

Replaced by: “(annual global production of about 400 Mt and forecasted to double by 2040) has no immediate replacement [12].”. Brand new reference Pathak, P., Sharma, S., & Ramakrishna, S. (2023). Circular transformation in plastic management lessens the carbon footprint of the plastic industry. Materials Today Sustainability22, 100365. added.

  1. What are the primary barriers to replacing fossil-based plastics with biodegradable alternatives?

We added the following information: “Widespread adaptation of natural alternative are currently impeded by higher production costs, insecure supply chains for raw materials, or cumbersome downstream processing; however, natural polymers as solutions exist or are in development that use inexpensive raw materials, ecologically friendly polymer production and extraction methods, in addition to being circular and sustainable from an end-of-life perspective [20]. “

  1. Could you comment on the functionality and longevity of PHA-based materials in comparison with fossil plastics?

We added: PHA is suitable for replacing fossil plastics in packaging, personal care and agriculture. There is a significant effort underway to use PHA in single-use packaging, where recycling packaging materials is difficult due to their complex multi-material construction and light weight. In foodservice packaging contamination of the packaging with food makes recycling even more difficult. Short shelf life (days, weeks, or months) foodservice packaging applications are ideal for a biodegradable and compostable material like PHA. Their excellent barrier properties to gases, moisture and fats and oils add to the advantages they offer in these applications.

However, PHA can also be used is long shelf-life uses such as in personal care packaging for soap and shampoo bottles that requires longer product shelf life. It must be emphasized that in long shelf life packaging and other durable uses PHA does not biodegrade and keeps their integrity and functionality just like fossil plastic bottles. Biodegradation starts when PHA comes in contact for long periods of time with adequate microflora, e.g., in composting facilities, in soil, or in water..

  1. Regarding the Green Chemistry Principle No 2 Atomic economy, it does not fit into the context of PHA. How would you compare the impact of CO2 emission by comparing with the synthesis of fossil plastics?

We changed accordingly to “Not applicable in the context of PHA. As natural, aerobic process, PHA biosynthesis creates intrinsically CO2 as side product; however, generated CO2 is embedded into the natural carbon cycle, in contrast to CO2 stemming from petrochemical products.“

  1. I would suggest shortening the conclusions part and moving tables 1 and 2 to the discussion part.

Conclusion part is now shortened, and the Tables are moved.

  1. The issues of fossil plastics consumption and pollution are complex and require comprehensive solutions that involve scientific advancements, policy changes, and societal shifts. Future direction’s part should be added with recommendations for future research.

We added: “Accelerating the transition to biological alternatives like PHA in substantial scale require a paradigm shift in many areas. They encompass availability of a variety of non-food renewable carbon sources and ecologically benign biopolymer recovery methods, designing materials that match existing fossil plastics in their properties and functionality, enabling their processing on existing infrastructure with little to no changes, designing waste management systems that improve their collection and sorting, and accepting and improving industrial composting and making them equally important to mechanical recycling. In addition, appropriate legislation to promote their use needs to focus on circularity and sustainability – renewability, recyclability, compostability and biodegradability. They must not focus on redefining plastics, natural polymers and other clearly established words and phrases. The FMCG (Fast-Moving-Consumer-Goods) and CPG (Consumer-Packaged-Goods) space, especially in foodservice, personal care, and health and beauty are ideally suited for such a transition to biological alternatives like PHA [74].

Finally, the overall cost of using fossil plastics that includes their negative environmental consequences needs to be holistically estimated beyond the cost of their fossil raw materials. This would allow for a true understanding and comparison of costs of fossil plastics with their biological and circular alternatives.

Finally, governments need to provide incentives in the form of direct support for more basic and applied research and development including the scaling newer technologies to enable the transition. This area represents the “New Economy” also called the “Bio-Economy” allowing local supply chains for materials, and new and innovative manufacturing processes that create significant new employment opportunities.”

  1. In technical and scientific writing, the use exclamation mark should be used only to end warning or caution statements or as specialized scientific notation. For other purposes, I would suggest using a period or question mark.

We fully agree; no more exclamation marks in the text!